# Oral Administration of Artemisone for the Treatment of Schistosomiasis: Formulation Challenges and In Vivo Efficacy

**DOI:** 10.3390/pharmaceutics12060509

**Published:** 2020-06-03

**Authors:** Johanna Zech, Daniel Gold, Nadeen Salaymeh, Netanel Cohen Sasson, Ithai Rabinowitch, Jacob Golenser, Karsten Mäder

**Affiliations:** 1Institute of Pharmacy, Martin Luther University Halle-Wittenberg, Kurt-Mothes-Str. 3, 06120 Halle (Saale), Germany; johanna.zech@pharmazie.uni-halle.de; 2Department of Clinical Microbiology and Immunology, Sackler Faculty of Medicine, Tel Aviv University, Ramat Aviv, Tel Aviv 69978, Israel; goldy@tauex.tau.ac.il; 3Department of Microbiology and Molecular Genetics, The Kuvin Centre for the Study of Infectious and Tropical Diseases, The Hebrew University of Jerusalem, Jerusalem 9112002, Israel; nadeen.salaymeh@mail.huji.ac.il; 4Department of Medical Neurobiology, Institute for Medical Research-Israel-Canada, Faculty of Medicine, Hebrew University of Jerusalem, Jerusalem 9112002, Israel; netanel.cohen1@mail.huji.ac.il (N.C.S.); ithai.rabinowitch@mail.huji.ac.il (I.R.)

**Keywords:** artemisone, schistosomiasis, lipid, self-microemulsifying, oral, SMEDDS, SNEDDS, microemulsion

## Abstract

Artemisone is an innovative artemisinin derivative with applications in the treatment of malaria, schistosomiasis and other diseases. However, its low aqueous solubility and tendency to degrade after solubilisation limits the translation of this drug into clinical practice. We developed a self-microemulsifying drug delivery system (SMEDDS), which is easy to produce (simple mixing) with a high drug load. In addition to known pharmaceutical excipients (Capmul MCM, Kolliphor HS15, propylene glycol), we identified Polysorb ID 46 as a beneficial new additional excipient. The physicochemical properties were characterized by dynamic light scattering, conductivity measurements, rheology and electron microscopy. High storage stability, even at 30 °C, was achieved. The orally administrated artemisone SMEDDS formulation was highly active in vivo in *S. mansoni* infected mice. Thorough elimination of the adult worms, their eggs and prevention of the deleterious granuloma formation in the livers of infected mice was observed even at a relatively low dose of the drug. The new formulation has a high potential to accelerate the clinical use of artemisone in schistosomiasis and malaria.

## 1. Introduction

Schistosomiasis is a chronic, parasitic disease transmitted by trematode worms of the genus *Schistosoma*. According to the UNHCO, over 250 million people are infected globally, and approximately 700 million are at risk in 74 countries where the infection is endemic. In sub-Saharan Africa, the disease causes over 200,000 deaths per year [1]. The current chemotherapy and prevention of schistosomiasis rely on two drugs only, oxamniquine and praziquantel. Oxamniquine is a prodrug which is only used against *S. mansoni* in cases of failure of the therapy with praziquantel. Oxamniquine is active, owing to DNA binding, but it affects only mature parasites and it occasionally provokes serious side effects. Resistance of *S. mansoni* to oxamniquine has been demonstrated both in the laboratory and in the field [2]. Praziquantel is active due to tegumental damage caused by calcium dyshomeostasis [3]. It is the first-line treatment, but unfortunately, it is also potent only against the late developing stages of the worm [3]. Moreover, recently there are increasing numbers of reports indicating non-successful praziquantel treatment because of induction of resistance [4,5,6].

Artemisinins are used as front-line drugs for the treatment of malaria. It was speculated that they are also active in schistosomes, as in plasmodia, by heme-initiated formation of free radicals [5,7,8]. Therefore, various artemisinin derivatives have been suggested as an alternative to praziquantel for the treatment of schistosomiasis. The different mechanisms of action of the artemisinins and praziquantel have led to numerous trials of their use as a combination therapy [8,9,10]. Most of the experiments reviewed in these papers demonstrate significant success of oral treatment in mice, typically using 100–400 mg/kg artemisinin derivatives. The reduction of worm numbers, egg burden and granuloma counts was dependent on drug concentration and treatment frequency. Lower doses of 10–30 mg/kg were needed for successful treatment of larger animals (e.g., hamsters, rabbits and dogs) [8]. Artemisinin-based drugs (mostly artesunate) were compared to praziquantel as monotherapy for human schistosomiasis. Patients treated with artesunate alone had significantly lower cure rates than those treated with praziquantel [11]. The authors suggest that the reason for this result is the fact that artesunate is active only against the early stages of the disease. This assumption would not be valid for two newly synthesized artemisinin derivatives [12] and artemisone (Figure 1A) [5], as they also affect late developmental stages. In addition to this advantage, artemisone is also known to be less neurotoxic than other artemisinins [13] and literature data indicate possible applications in the treatment of cancer [14], bovine diseases [15] and viral infections [16] besides its use in malaria. Therefore, the development of a safe, efficacious and easily administered formulation of artemisone is not only important for the treatment of schistosomiasis, but it has also implications for other diseases.

In order to achieve a therapeutic effect, sufficient concentrations of artemisone must be present in the body over longer time periods (days-weeks). Both oral and parenteral administration may be considered to achieve this goal. Overdosing has to be circumvented in both cases because artemisinins are toxic at high concentrations. So far, almost all commercial parenteral controlled release systems are based on polylactide (PLA) or poly-(lactide-co-glycolide) (PLGA) polymers. These polymers often undergo autocatalytic degradation, which leads to the formation of acidic microenvironments, complex and nonlinear release profiles and potential degradation of the drug prior to its release. Recently, new polymeric systems have been developed, which, after subcutaneous injection, released artemisone slowly and eliminated most of the schistosomes [4]. The advantage of parenteral administration is the avoidance of the intestinal absorption process. Compared to the oral administration, the disadvantages of the parenteral route include the invasiveness of administration, the requirement for sterility, higher risk of administration induced infections and, last but not least, higher production costs. Disadvantages of the oral route are the higher frequency of administration and the high variability of the amount which is absorbed. In the case of artemisone, poor oral absorption required extremely high doses and resulted in variable bioavailability. Despite this, oral administration of the drug would still be preferable, especially when considering the precarious health care conditions prevalent in many of the regions where *S. mansoni* is encountered. Therefore, it is a rational and attractive goal to decrease the requisite administered dose by increasing the absorption efficacy. This will also lead to a decrease in variability and the overall costs of the treatment.

Artemisone has a low aqueous solubility (89 mg/L H_2_O) and lipophilic properties (logP = 2.49) [13,17]. The lipophilicity of the molecule is also apparent by the fact that artemisone accumulates within *P. falciparum’s* neutral lipid compartments [18]. Therefore, lipid formulations of artemisone and other artemisinins are potentially beneficial for the formulation of these compounds. Burger et al. developed classical o/w-nanoemulsions for the topical use of artemisone that were successfully examined in vitro [19]. Glycerol monostearate or Span60-based nanodispersions have also been described and tested in vitro [14]. Other surfactant-based nanodispersions were unable to deliver the drug into the skin in vitro [20]. Oral formulations taking advantage of the Pheroid™ technology were investigated in mice [17], demonstrating a 5.57-fold higher absorption of artemisone compared to an aqueous vehicle containing 10% DMSO. The Pheroid™ technology is based on Cremophor™ EL stabilized o/w-dispersions (size in the lower micron range) of vitamin F esters. Vitamin F consists mainly of multiple unsaturated fatty acids (e.g., linoleic and linolenic acid), which are sensitive to oxidation. Purging with N_2_O and storage in glass vials is required to reduce the oxygen-induced oxidation of the excipients. The intrinsic susceptibility to oxidation is a clear disadvantage of the Pheroid™ technology because it requires special filling, shipping and storage conditions.

Unfortunately, neither chemical nor physical (size increase, phase separation) stability data have been published for artemisone formulations. To date, all formulations published consist of aqueous dispersions. In previous studies, we investigated the impact of water and the pH value on the stability of artemisone [21]. At pH 3 a complete degradation was observed within one week. The degradation rate was slower at higher pH values, although only 20 and 75% of the intact compound remained after one month at pH 5 and pH 7.4, respectively (Figure 1B). Therefore, the preparation of a non-aqueous formulation or a formulation with low water activity would be desirable to achieve sufficient stability of the drug.

Challenges for the development of an oral artemisone formulation include (1) stability problems (endoperoxide structure: sensitivity to traces of metal ions, water and light) and (2) variability of oral absorption due to the poor aqueous solubility. Previously published artemisone formulations for oral and topical application have not been able to successfully address these challenges, leading to:Chemical and thermodynamic instability due to the use of aqueous dispersionsLow drug load (the major volume is water)Use of deleterious organic solventsDifficult to scale-up technologies (ultrasonication)Special requirements/high sensitivity to oxidation (for example in Pheroid™ technology)

It was therefore our aim to develop an oral formulation which avoids these limitations and could potentially facilitate the translation of artemisone into clinical practice. Consequently, the desired properties of the artemisone formulation we strove to achieve include:High drug load and in vivo efficacyEasy manufacturing process without special equipmentHigh storage stability (thermodynamic and chemical)High potential for translation into clinics

This would then enable us to test the use of artemisone in this particular formulation for its efficacy in the treatment of schistosomiasis in a *S. mansoni* mouse model safely and effectively. Because of the lipophilic properties of artemisone, we developed a self-emulsifying drug delivery system (SMEDDS). SMEDDS are generally made from mixtures of lipophilic, water-insoluble excipients (mono-, di- and triglycerides, HLB < 8) with water-soluble emulsifiers (typical HLB range 14–16) and cosurfactants (e.g., ethanol or propylene glycol). To decrease the danger of oxidation problems common with lipids, we avoided unsaturated excipients. In addition to using well-established excipients for the formulation (e.g., medium chain partial glycerides and triglycerides, PEGylated surfactants), we selected Polysorb ID 46 as a novel excipient candidate. Polysorb ID 46 is an isosorbide diester with medium chain fatty acids (C8–C10) which was developed as a bio-based, “green” plasticizer to replace phthalates in non-pharma applications (e.g., PVC, plastic toys). However, Polysorb ID 46 is also a promising candidate as a pharmaceutical excipient because it has good solvent properties for poorly soluble drugs [22] and a very low toxicity, as tested in mice and rats [23,24]. The NOAEL value (No Observed Adverse Effect Level) for the oral administration to rats is 2000 mg/kg bw/day, and the oral administration of the substance can be regarded as “practically nontoxic” [24]. The formulation development included the following steps:Pre-screening of drug solubility in different excipients.Selection of components by ternary phase diagrams.Optimization of the composition based on physicochemical properties.Stability studies.In vitro activity.In vivo activity.

The formulation was characterized by dynamic light scattering, light microscopy, electron microscopy, conductivity measurements, monitoring of drug stability, rheology, dilution behavior, sensitivity to different pH values and salinity. An in vivo proof of concept for anti-schistosomal effect was decisive. Overall, we present the successful development of novel artemisone SMEDDS formulation that could be translated into clinical use.

## 2. Materials and Methods

### 2.1. Materials

Artemisone, (4-[(1R,4S,5R,8S,9R,10R,12R,13R)-1,5,9-trimethyl-11,14,15,16-tetraoxatetracyclo- hexadecan-10-yl]-1,4-thiazinane1,1-dioxide) [ART] was kindly donated by Cipla (Mumbai, India) and the laboratory of Richard K. Haynes at North-West University (Potchefstroom, South Africa). Capmul^®^ MCM EP (Glycerol monocaprylocaprate) and Captex 355^®^ EP/NF (medium chain triglycerides/triglycerides of caprylic/capric acid) were supplied by Abitec (Columbus, OH, USA). Polysorb^®^ ID 46 (isosorbide caprylocaprate diester) was gifted by Roquette (Lestrem, France) and Kolliphor^®^ HS15 (polyoxyl (15) hydroxystearate) by BASF (Ludwigshafen, Germany). The semisolid excipient Kolliphor^®^ HS 15 was molten before use to ensure a homogenous composition. Propylene glycol was bought from Caesar & Loretz GmbH (Hilden, Germany). Any HPLC solvents were purchased as HPLC grade; all other chemicals were of analytical grade. Only double distilled water was used in the experiments. PBS pH 7.4 for physicochemical characterization of the formulation was prepared according to Ph. Eur. and preserved with 0.02% NaN_3_. For biological determinations, we used PBS without NaN_3_ (Biological Industries, Bet Haemek, Israel).

All % are given as weight % unless stated otherwise.

### 2.2. Methods

#### 2.2.1. Formulation Development

##### Solubility of Artemisone in Organic Solvents, Excipients and SMEDDS

The solubility of artemisone in organic solvents and lipid excipients was determined according to the following procedure: 200 mg artemisone/1.0 mL excipient were placed on an orbital mixing heating plate (Torrey Pines Scientific, Carlsbad, CA, USA) at 25 °C for 24 h under light exclusion. Samples were centrifuged and the supernatant filtered and diluted with ethanol as necessary before HPLC analysis. Samples were prepared in triplicates.

Artemisone was quantified with high-performance liquid chromatography (HPLC) (Agilent/HP 1100 HPLC series, high-pressure pump, degasser, UV-VIS detector and autosampler, now Agilent Technologies, Inc., Santa Clara, CA, USA), using a method we developed and validated according to the ICH guidelines [25], over a range of 10–500 µg artemisone/mL (Table 1).

Experiments were performed using an Eurospher 100-5 C18 100 × 2 mm column (Knauer, Berlin, Germany). Dilutions of artemisone in ethanol were used as standards to validate the method. A linear calibration curve was obtained (*R*^2^ = 0.9997; DL = 0.44 µg/mL, LQ = 1.35 µg/mL). Recovery of artemisone was determined from the parent, water-free formulation SMEDDS-100 (50 mg/g ART), and PBS dilutions of SMEDDS-100 (SMEDDS-100 50% and 20% in PBS). The recovery rates for the SMEDDS-100, SMEDDS-50 (25 mg/g ART) and SMEDDS-20 (10 mg/g ART) were determined from the samples after dilution with ethanol. High recovery rates of (97.4 +/− 1.1)% (SMEDDS-100); (100.6 +/− 1.9)% (SMEDDS-50) and (99.1 +/− 1.4)% (SMEDDS-20) were found. For 80 µg/mL artemisone in PBS pH 6.5, samples were measured undiluted and recovery was (96.2 +/− 3.4)%.

##### Ternary Phase Diagrams

All formulation components (lipids, surfactant, co-surfactant) with the exception of water were blended by vortex for 20 s at 2500 rpm. Water was added stepwise and the samples mixed for 20 s after each step before optical evaluation (turbid/clear/opalescent). Experiments were performed at 25 °C.

##### Formulation Preparation

For the formulation of the SMEDDS-100, Kolliphor^®^ HS 15 was melted and combined with propylene glycol, Polysorb^®^ ID 46 and Capmul^®^ MCM by stirring on a magnetic stirring plate until a homogenous, clear and viscous liquid was obtained. To prepare the PBS dilutions of SMEDDS-100, this formulation was mixed with PBS at 1:1 and 1:4 ratios, leading to the formulations SMEDDS-50 and SMEDDS-20.

##### Assessment of Self-Microemulsifying Potential of Lipid Formulations

SMEDDS-100, traced with the lipophilic dye Sudan Red G to ease visibility, was placed in glass vials and carefully overlaid with 1 or 4 times the amount of PBS or bi-distilled water. Samples were shaken on an orbital mixing heating plate (Torrey Pines Scientific) at 25 °C or 37 °C and were assessed every 10 min for the dispersion degree. In addition, the self-emulsifying properties of the lipid formulation were also evaluated via polarized light microscopy (Zeiss Axiolab, Oberkochen, Germany) on a tempered objective slide by placing a drop of lipid formulation on a PBS surface.

##### Stability of Artemisone in SMEDDS and PBS

Artemisone (50 mg/g) was dissolved in SMEDDS-100. For SMEDDS-50-ART and SMEDDS-20-ART, PBS was added to the drug-loaded formulation to give the microemulsions, containing 25 mg/g and 10 mg/g artemisone, respectively. A solution of 80 µg/mL artemisone in PBS pH 6.5 Ph.Eur. was prepared and used as control, the pH found for SMEDDS-50 and SMEDDS-20 also being around 6.5. Triplicates of all three preparations were stored in glass vials with Teflon sealed lids at 30 °C under light exclusion. Samples were taken after 14, 28, 42, 56, and 92 days and analyzed via HPLC to monitor drug stability.

##### Stability at Different Temperatures and Ionic Strength

To assess thermal stability and temperature-mediated changes in the microstructure of the SMEDDS and microemulsions, we used conductivity measurements, DLS and optically visible changes. The term “cloud point” is applied here to determine the temperature at which the clear-opalescent formulation turned opaque-white.

The effect of ionic strength on the stability and microstructure of the formulations were evaluated by using double distilled water, 5-fold concentrated PBS and 10-fold concentrated PBS instead of the regular buffer to prepare SMEDDS-20 and SMEDDS-50. Results were compared to the samples prepared with PBS and the cloud point was determined for all combinations. DLS measurements of all the formulations were performed.

##### Rheology

Density and viscosity were assessed using a Westphal balance and capillary viscometers (SCHOTT Instruments GmbH, Mainz, Germany) in a constant temperature bath. The Westphal balance allows determining the density of liquids through their buoyant force compared to water, measured by immersion of a glass float of specific volume in that liquid. Dynamic viscosity was calculated from flow time in the capillary viscometer and density. Rheology measurements were carried out with a Kinexus lab+ rotational rheometer (Malvern Panalytical, Kassel, Germany) equipped with a cone-plate geometry (diameter 60 mm, angle 1°, gap 0.03 mm), heating unit and solvent trap. Samples of 1 mL were analyzed over a frequency range of 0.1–1000 s^−1^ (10/decade) over 30 min at 25 °C.

##### Dynamic Light Scattering (DLS)

Analysis was performed with the Zetasizer Nano ZS (Malvern Instruments, Malvern, UK) using Non-Invasive Back Scatter (NIBS) with detection at 173°. Equilibration time was set to 10 min for 25 °C and 15 min for 37 °C. Three measurements per sample were taken. To investigate the impact of temperature, samples were heated in steps of 2 °C from 25 °C to 55 °C and equilibrated for 5 min before each measurement.

##### Cryogenic Electron Microscopy

Transmission electron microscopy was used for SMEDDS-20. Samples were prepared on copper grids, frozen in liquid propane and then stored in liquid nitrogen. For SMEDDS-50, freeze-fracture electron microscopy was performed. Cryofixation was done on gold grids and samples stored and fractured in liquid nitrogen. Replicas were created by coating with platinum (2 nm, 45°) and carbon (2 nm, 180°), consequent treatment with hypochlorite and washing with acetone and water. Electron microscopy was performed with a Libra 120 Plus transmission electron microscope, equipped with a 2k Slow Scan CCD camera (Zeiss, Oberkochen, Germany).

#### 2.2.2. Measurement of Biological Activity in Culture

##### *Caenorhabditis elegans* 

*C. elegans* served as a control non-pathogenic helminth that does not feed on erythrocytes. *C. elegans* Bristol strain N2 worms were grown and maintained under standard conditions at 20 °C on 6 cm agar plates containing 5 mL nematode growth medium (NGM) in 2% agar. Plates were seeded with 150 µL *Escherichia coli* strain OP50 grown overnight from a single colony in 20 mL LB.

Similar plates were used as test plates. 500 µL of artemisone in SMEDDS-20 were applied to each test plate and allowed to absorb in the agar for two hours. Two age-synchronized young adult worms (that have just reached L4 stage), three days post-hatching, were transferred to a test plate. The worms were left on the plate to lay eggs for 24 h and were then removed. The eggs in each plate were counted. After an additional 48 h, the number of adult worms in each plate was determined. *C. elegans* eggs and adult worms were counted using a stereomicroscope.

##### *Plasmodium falciparum* 

*P. falciparum* NF54 parasites were cultivated at 4% hematocrit in RPMI 1640 medium, 0.5% Albumax II (Invitrogen, Carlsbad, CA, USA), 0.25% sodium bicarbonate, and 0.1 mg/mL gentamicin. 200 µL of parasite suspension at 1% parasitemia were incubated in a Nunc microplate, in 96 flat-bottom wells, at 37 °C, in an atmosphere of 5% oxygen, 5% carbon dioxide and 90% nitrogen. Drug dilutions were added in triplicates in 25 µL. After 48 h *P. falciparum* parasitaemias were estimated by microscopic determination of the percent of parasitized erythrocytes in a total of 5000 erythrocytes, using Giemsa stained blood smears that were prepared from triplicate individual wells.

#### 2.2.3. Measurement of Biological Activity in Vivo

##### *S. mansoni* 

Experiments were performed using the Puerto Rican isolate obtained from the National Institute of Allergy and Infectious Diseases, National Institute of Health (Bethesda, MD, USA). The life cycle of *S. mansoni* was maintained in ICR mice and *Biomphalaria glabrata* snails. The snails were raised and kept at 26 °C in aerated aquaria. Mice were routinely infected by subcutaneous injection of about 200 cercariae each. Seven to eight weeks post-infection, schistosome eggs were extracted from the livers, isolated and hatched in water to obtain miracidia. Snails were infected individually by exposure to 7–8 miracidia each. Four weeks later the snails started to shed cercariae. The number of cercariae was adjusted to aliquots of 200/300 µL sterile tap water that were injected subcutaneously into each mouse.

##### Mice

Male ICR mice were purchased from Harlan Laboratories (Rehovot, Israel). These mice were used for the *Schistosoma* infections at the Tel-Aviv University (Tel Aviv University ethical committee number 01-13-076). The mice were infected a few days later by 150 cercariae.

##### Treatment of *Schistosoma* Infected Mice

ICR male mice, 7–8 weeks old, ~33 g at infection, were treated by gavage 40 mg/kg, twice a day on days 23–25 and 29–31 post-infection. The drug was given in the form of 300 µL of SMEDDS-20-ART.

##### Assessments of Treatments

*Schistosoma* numbers were determined by counting worms from dissected and squashed livers and mesenteric veins of dissected intestines of the mice 49–51 days post-infection. Liver samples of mice were formalin-fixed 4% (v/v) and paraffin-embedded, 4-micron sections were cut and stained with H&E to assess egg and granuloma density. Granulomas and individual eggs were counted for each liver section and the areas of the liver sections were measured.

## 3. Results

### 3.1. Formulation Development

In the first step, the solubility of artemisone was measured in different organic solvents and various lipid excipients. A very high solubility (>100 mg/mL) was found for acetone, DMF, dichloromethane and DMSO. Artemisone was less soluble in ethanol (30 mg/mL), methanol (50 mg/mL), propylene glycol (8.5 mg/mL), and hexane (0.8 mg/mL). The solubility of artemisone in medium chain triglycerides (MCT) was 22.5 mg/mL. An almost three times higher solubility (62.5 mg/mL) was observed in medium chain mono- and diglycerides (Capmul MCM), which are, compared to MCT, more polar. Polysorb ID 46 was also identified as a good solvent for artemisone with a solubility of 55.0 mg/mL. MCT, Capmul MCM and Polysorb ID 46 were selected as lipophilic components because they show high solubilization capacity and do not contain unsaturated fatty acids. In addition to lipophilic excipients, the formation of a SMEDDS requires further components, which enable the self-emulsifying properties. Kolliphor HS15 and propylene glycol (ratio 5 + 1) were selected as surfactant and co-surfactant. Kolliphor HS15 was chosen because it does not contain unsaturated fatty acids and it is a particularly useful excipient for the development of self-emulsifying systems [26,27].

The performance of different mixtures was tested in screening experiments. As a result, Polysorb ID 46 mixtures with MCT (1:1) and MCM (1:1 and 2:1) were the most encouraging mixtures for the lipophilic phase. Therefore, in the next step, detailed pseudo-ternary phase diagrams of different ratios of (I) surfactant/cosurfactant, (II) lipid components and (III) water were recorded (Figure 2).

A robust SMEDDS system should show a high area of isotropic regions which extend into the corner with high water contents, reflecting the dilution behavior which will take place in vivo. Clear, liquid formulations considered as ME could be obtained for all three systems.

Considering the solubilization capacity of the components for artemisone and the expected dilution behavior which was concluded from the phase diagrams, we selected the following formulation for further investigations: 40% Capmul MCM/Polysorb ID 46 1:1 mixture as lipophilic phase and 60% of a Kolliphor HS15/propylene glycol (5 + 1) mixture. The formulation is called SMEDDS-100 and marked with a red triangle in Figure 2. The dilution pathway of SMEDDS-100 is indicated as a dotted line.

Phase diagrams do not always predict the self-emulsification process correctly. For example, highly viscous liquid crystalline phases may be formed at the SMEDDS/water interface, which will slow, delay or even prevent self-emulsification. Therefore, the self-microemulsification properties of SMEDDS-100 were tested in PBS and bi-distilled water. As shown in Figure 3, self-microemulsification was observed in both media.

Polarized light microscopy investigations of the SMEDDS-100 interface with PBS indicated the formation of an ordered lyotropic liquid crystalline phase that disappeared with further mixing of the two components (Figure 4).

We selected the non-aqueous system SMEDDS-100, and two dilutions of SMEDDS-100 with PBS at ratios of 1:1 (SMEDDS-50) and 1:4 (SMEDDS-20) for further development. The selected formulations are indicated by triangles in Figure 2 and their detailed compositions are given in Table 2.

In a next step, the saturation solubility of artemisone in the selected formulations was determined. SMEDDS-100, SMEDDS-50 and SMEDDS-20 were able to dissolve 58.9 mg/g, 15.5 mg/g and 5.5 mg/g of artemisone, respectively. With respect to the low solubility of artemisone in water, (89 µg/mL H_2_O), the non-aqueous SMEDDS-100 increased the solubility approximately 660-fold. Even the formulation SMEDDS-20, which contains 80% water, solubilized artemisone more than 60-fold compared to PBS.

#### 3.1.1. Physical Stability

To gain further insights into the properties of the SMEDDS as a function of the PBS content, additional characterization studies were performed. At low water contents (10–30 wt. % water), the turbidity of the formulations indicated the formation of w/o-emulsions. However, all other formulations were not turbid. Formulations with a PBS content of 80 and 90% showed a strong blue opalescence (Appendix A). This behavior was identical for both water and PBS as diluent. Formulations with a water content of ≤70% were able to dissolve Sudan red powder immediately, indicating the presence of a percolating lipophilic phase (Appendix A). Both the composition and the temperature had an impact on the particles size determined by dynamic light scattering (Appendix A). In general, higher water content and increased temperature increased the particle size and polydispersity. At room temperature samples with low water content showed unimodal distributions in the range of about 20–30 nm. Because microemulsions are highly dynamic systems, fast fluctuations of association processes contribute to the measured particle size. More detailed investigations on the impact of temperature and drug load on conductivity and particle size were performed for SMEDDS-50 and SMEDDS-20 (Appendix A). The results indicate (a) a higher impact and more complex behavior of temperature-induced changes for the SMEDDS-20 system (Appendix A top) and (b) drug load-induced changes were not observed at 25 °C, although a moderate increase in particle size was observed at 37 °C (Appendix A bottom). Further experiments were conducted to study the impact of ionic strength on the cloud point. The results indicate that higher ionic strength decreases the cloud point, but non-physiological values would be required to drop it below body temperature (Appendix A).

The rheological properties are important for processing (e.g., capsule filling) and administration. Especially for preclinical trials in mice, the oral administration by gavage will be easier to conduct and more reproducible with low-viscous fluids. The dynamic viscosity determined by capillary rheology for SMEDDS-100 was (142 ± 0.1) mPas at 25 °C and (77 ± 1) mPas at 37 °C. For SMEDDS-50, slightly lower values were found ((118 ± 4) mPas at 25 °C and (66 ± 0.1) mPas at 37 °C) (Appendix A top). SMEDDS-20 with a water content of 80% was less viscous (<30 mPas) at both temperatures. These values were confirmed by measurements with a rotational rheometer (Appendix A bottom) which indicated an almost shear-independent behavior of SMEDDS-50 (130 mPas) and a shear-dependent, thixotropic behavior of the SMEDDS-20 formulation (initial viscosity 8 mPas). In addition, cryogenic electron microscopy was performed to obtain more knowledge about the structure of the SMEDDS. The pictures indicate a bicontinuous structure for SMEDDS-50 and cylindrical micelles for SMEDDS-20 (Appendix A).

#### 3.1.2. Storage Stability

As mentioned in the introduction, artemisone often has a low storage stability in solution. Therefore, the stability of artemisone in SMEDDS-100, SMEDDS-20 and SMEDDS-50 was analyzed via HPLC and compared to the compound stability in PBS pH 6.5. The samples were stored at 30 °C to reflect the expected conditions typical of its later use, where storage conditions under refrigeration might be unrealistic. The SMEDDS-20 showed crystallization occurring within one week. While only 51.5 ± 0.4% of artemisone was recovered in PBS pH 6.5 after three months, 98.5 ± 0.9 and 98.5 ± 1.3% of the drug were recovered in SMEDDS-100 and SMEDDS-50 formulations, respectively (Figure 5).

This remarkably high stability of artemisone even in the presence of 50% water suggests that the drug is localized within the lipophilic domains of the microemulsion which, together with the interfacial layer of surfactant, afforded protection from degradation [28,29]. Such protection allows storage of the drug even in the presence of water for several weeks and adds to the integrity of artemisone in the GI tract after oral application.

To summarize key findings of the formulation development process, self-emulsifying artemisone formulations could be developed with a simple manufacturing process (mixing), high drug loading capacity and a high storage stability even at 30 °C. In the next step, initial biological activity experiments were performed to verify the therapeutic potential of the formulations.

### 3.2. Biological Activity 

#### 3.2.1. In Vitro Examinations

##### *P. Falciparum* 

Both the free drug and SMEDDS-20-ART were highly effective in vitro against the development of *P. falciparum*; in both cases the ED_50_ was 1–2 ng/mL (Figure 6). Parasitaemia was completely blocked at concentrations below 11 ng/mL. SMEDDS-20 alone had no effect at the examined concentrations.

##### *C. elegans* 

The number of eggs (85 ± 10) and adult worms (104 ± 5) in untreated cultures was not affected by SMEDDS-20 or by SMEDDS-20-ART in drug concentrations of up to 45 µg/mL. Because the worm eggs are very small, some of them were missed during the counting; the adult worms that developed from these eggs were detected later, hence the slight difference between egg and worm number. The in vitro examinations indicate a specific anti-schistosomal effect.

#### 3.2.2. In Vivo Examinations

Infected mice were treated by gavage twice a day, at days 23–25 and 29–31 post-infection. One group received 300 µL SMEDDS-20-ART, delivering a dose of artemisone of 40 mg/kg body weight. Another group received 300 µL of the drug-free SMEDDS-20 formulation (vehicle), a third group no treatment at all. There was no difference in *Schistosoma* counts between the untreated and the vehicle-treated mice. In contrast, treatment with SMEDDS-20-ART was highly effective (Figure 7): no schistosomes were detected in 2/6 of the treated mice, and a reduction of about 94% of the worms in 4/6 mice. The remaining few schistosomes that could be found in these mice were degenerative.

Liver histology of the vehicle group revealed a huge granulomatous reaction of 20–76 lesions per cm^2^ in response to 63–172 eggs/cm^2^ (Table 3). All eggs were in different stages of degeneration. Eggs were not seen in SMEDDS-ART-20-treated mice. There was no inflammatory response in most of the SMEDDS-ART-20-treated mice, but in 2/6 animals of this group there was still a background inflammatory response near liver sinusoids (Figure 8). Gross macroscopic observation revealed numerous lesions in livers of the vehicle-treated mice in contrast with the normal appearance of livers of the artemisone-treated mice (Figure 9).

## 4. Discussion

This paper presents a new efficient delivery method of artemisone for treating schistosomiasis in a mouse model. A SMEEDS was developed successfully with the following characteristics: (1) high drug load, (2) simple manufacturing process without special equipment, (3) high storage stability and (4) profound efficacy of oral administration of the dissolved drug, allowing administration of only one-tenth of the dose previously administered in suspensions used for the treatment of schistosomiasis [4]. The artemisone-loaded SMEDDS system, which was delivered to mice by gavage at an early stage of the infection before the worms reach adulthood, completely eliminated or killed the worms and prevented egg production, the main cause of pathology due to deleterious immune response to egg antigens [30,31].

Owing to a speculated identical mode of action of artemisinins against malaria and schistosomiasis (see introduction) various artemisinin derivatives have been examined for treating schistosomiasis. The most common mode of delivery was the oral route. Likewise, a single oral dose of artesunate (400 mg/kg in 3% ethanol and 7% Tween 80) given 21 days post-infection, yielded a 66% reduction of worm load [10]. A similar procedure using other artemisinin derivatives (artemiside, artemisone, CKWO3 and artemether) yielded more than 90% reduction [5]. Significant reduction of *S. mansoni* load was found in infected mice treated with new artemisinin derivatives, artesunic acid and dihydroartemisinin acetate [12]. In these experiments, the drugs were mixed with polyvinylpyrrolidone to increase solubility and oral treatment with 100 mg/kg was performed once, 21 or 45 or 60 days post-infection. It is likely that additional treatments in all the above-listed investigations would improve the results, but it can also be speculated that only a fraction of the dose reached the bloodstream following the gavage; otherwise it would induce various toxic effects that have been attributed to artemisinins [32,33]. The question of artemisone toxicity has been dealt with by Guiguemde et al. [34]. They conclude that the deleterious effects of artemisone do not exceed those of the commonly used artesunate. Anyhow, all artemisinins are not recommended for use during the first trimester of pregnancy [35].

In most of the artemisone-treated mice in our experiments there was a total elimination of the worms; although, in some, the histological sections showed a minor immune response that was attributed to a background inflammatory response [36]. This background inflammation is seen near veins and sinusoids, probably because of abundance of *Schistosoma* antigens following the treatment. The burst of antigens is a known phenomenon that may occur after treatment of schistosomiasis with anti-schistosomal drugs [37,38] or, similarly, filariasis treated with diethylcarbamazine [39]. In the placebo-treated mice there was a huge granulomatous inflammatory response that eventually led to mouse mortality (data not shown). A similar deleterious immune reaction is also the reason for morbidity and mortality in human infections [40].

The pathology caused by schistosomes in mice and in human patients might not be identical. However, so far, drugs that were active in mice were also active in humans. Moreover, in general, drug doses per weight are much smaller in humans than in the small rodents that are experimentally used [41]. Therefore, in human patients this might reduce potential toxicity that has been attributed to artemisinins. Our results suggest a possible alternative that together with praziquantel or other drugs will solve the pressing need for new drugs against schistosomiasis [42]. In summary, a new formulation of artemisone has been developed, which might help to transfer the high therapeutic potential of this molecule for several important diseases into clinical practice.

## Figures and Tables

**Figure 1 pharmaceutics-12-00509-f001:**
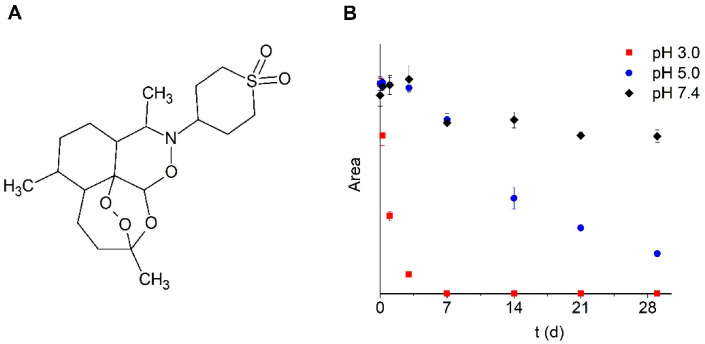
(**A**) Chemical structure of artemisone and (**B**) impact of pH on stability of artemisone in aqueous media, measured with HPLC.

**Figure 2 pharmaceutics-12-00509-f002:**
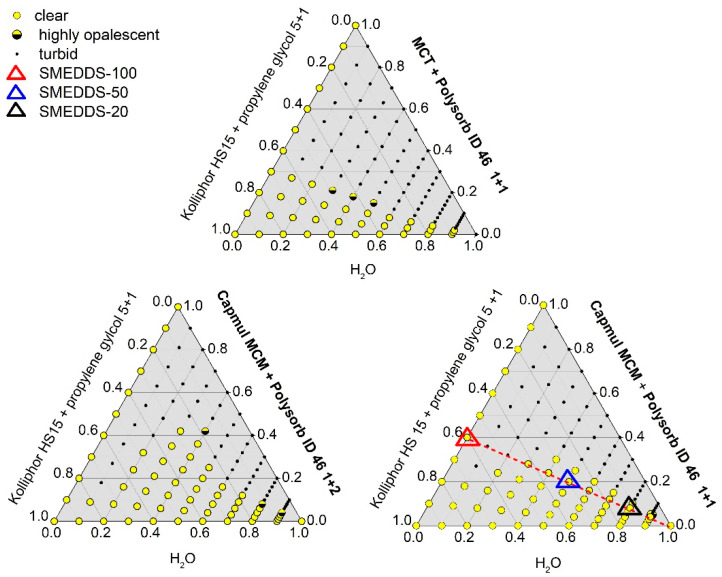
Impact of the composition of the lipid components on the phase behavior: ternary phase diagrams (25 °C) of pseudo ternary mixtures of (**I**) distilled water, (**II**) surfactant/co-surfactant and (**III**) lipid excipients. Any formulations that did not appear completely clear (ranging from slightly cloudy to milky white) were classified as turbid. Open triangles show the formulations SMEDDS-100; SMEDDS-50 and SMEDDS-20 which were selected for further detailed studies. The dotted line indicates the dilution pathway of the SMEDDS-100 formulation.

**Figure 3 pharmaceutics-12-00509-f003:**
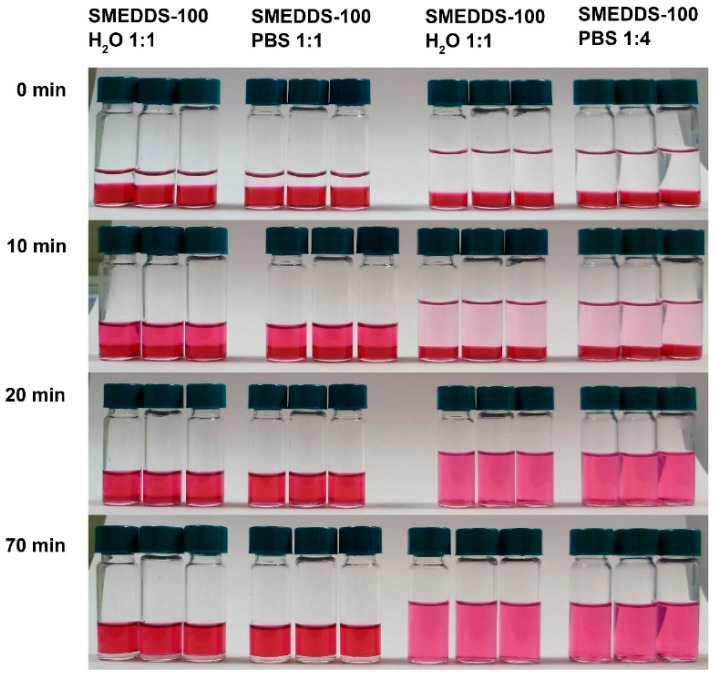
Self-microemulsifying properties of SMEDDS-100, dyed with Sudan Red, overlaid with distilled water or PBS at 1:1 or 1:4 ratios. The experiment was conducted with gentle shaking at 25 °C.

**Figure 4 pharmaceutics-12-00509-f004:**
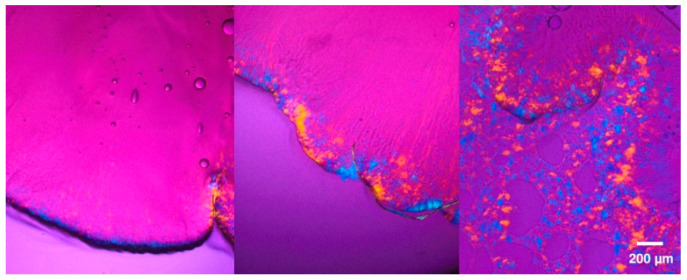
Polarized light microscopy of a lyotropic liquid crystalline phase, developing at the interface of SMEDDS (emerging from the top) and PBS when liquids are brought into contact. Photographs were taken over of the course of 30 s.

**Figure 5 pharmaceutics-12-00509-f005:**
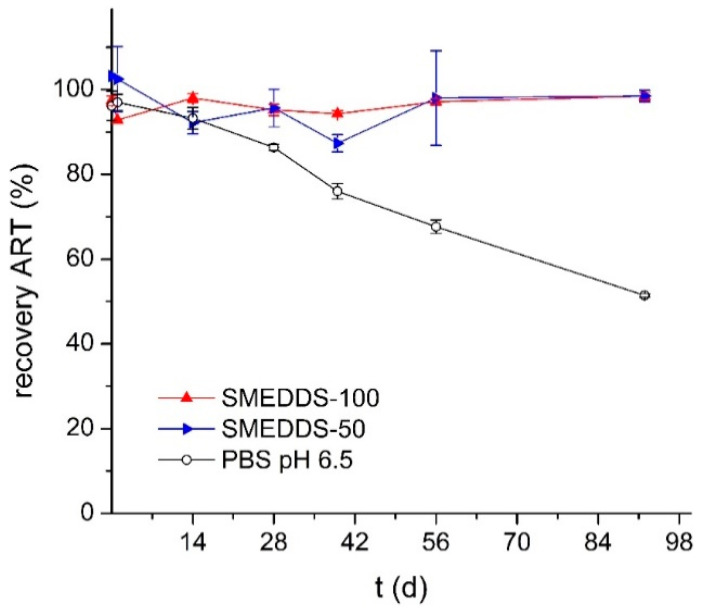
Stability of artemisone in SMEDDS-100, SMEDDS-50 and PBS pH 6.5 (pH equivalent to that of SMEDDS-50), stored at 30 °C under light exclusion. Drug content was measured by HPLC.

**Figure 6 pharmaceutics-12-00509-f006:**
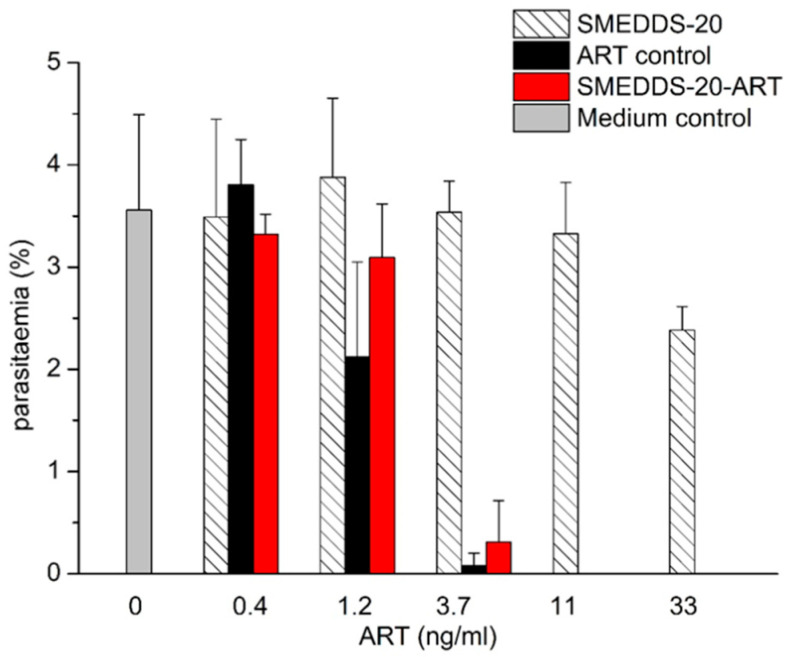
In vitro efficacy of free artemisone and SMEDDS-20-ART on *P. falciparum* in culture. Parasitaemia at time point zero was about 1%.

**Figure 7 pharmaceutics-12-00509-f007:**
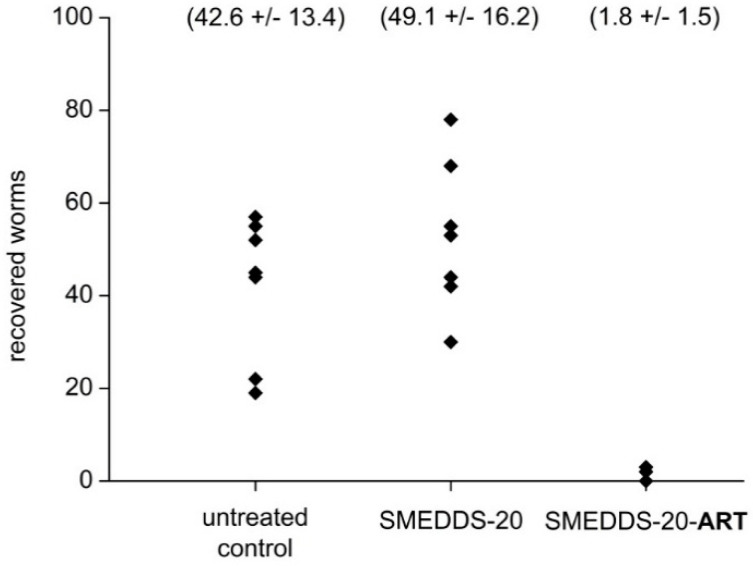
The effect of SMEDDS-20-ART and drug-free SMEDDS-20 administered by gavage on the number of worms in *S. mansoni* infected mice. Each point represents one mouse (*n* = 9 in the untreated and vehicle groups, *n* = 6 in the artemisone treated group).

**Figure 8 pharmaceutics-12-00509-f008:**
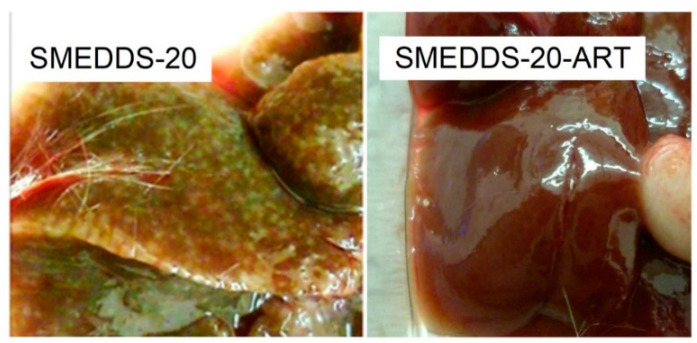
Gross macroscopic observation of livers of infected mice, vehicle or artemisone-treated.

**Figure 9 pharmaceutics-12-00509-f009:**
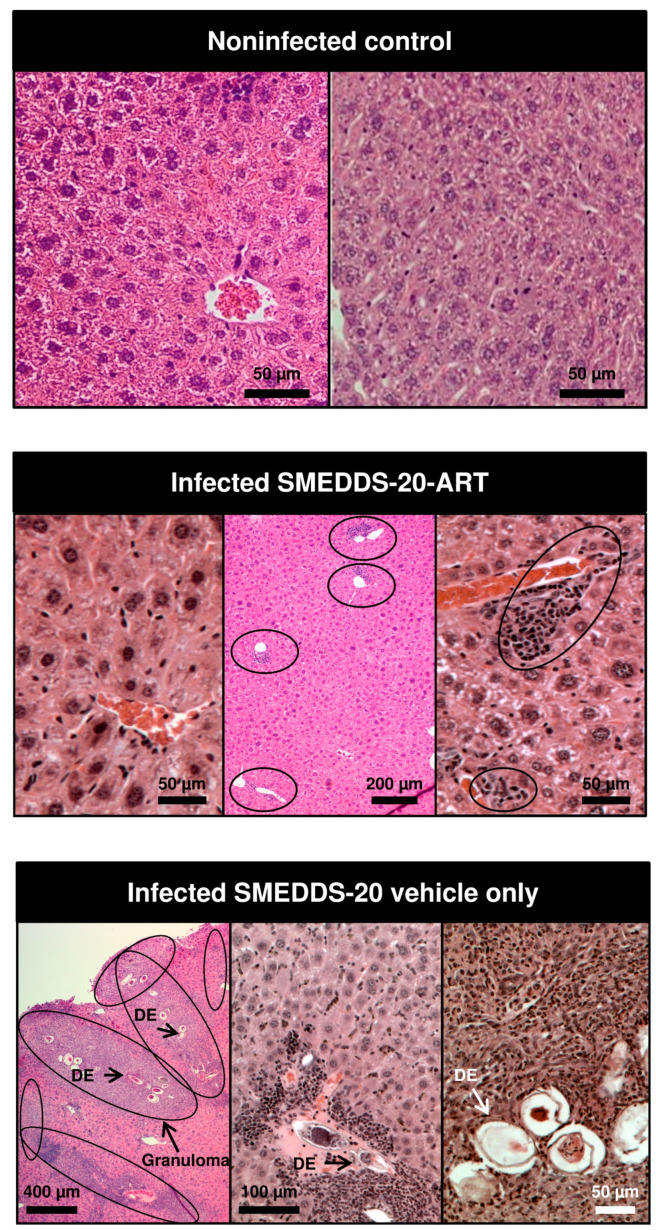
Liver histology, H&E stained, top: non-infected control; middle: artemisone-treated mice infected with *S. mansoni*. IC = inflammatory cells; bottom: infected mice treated with drug-free SMEDDS-20, DE = degenerated eggs.

**Table 1 pharmaceutics-12-00509-t001:** HPLC parameters for the quantitative determination of artemisone.

t [min]	Mobile Phase V/V/V			Flow Rate	Sample Injection Volume	T Column	UV Detection
	ACN	H_2_O	Formic Acid				
0–10 gradient	52.5→100	47.5→0	0.1	0.3 mL/min	10 µL	35 °C	200 nm
10–20 isocratic	100	0	0.1				
20–30 isocratic	52.5	47.5	0.1				

**Table 2 pharmaceutics-12-00509-t002:** Composition of selected formulation SMEDDS-100, SMEDDS-50 and SMEDDS-20.

Excipient	SMEDDS-100	SMEDDS-50	SMEDDS-20
Kolliphor^®^ HS15	50	25	10
Propylene glycol	10	5	2
Polysorb^®^ ID 46	20	10	4
Capmul^®^ MCM	20	10	4
PBS	0	50	80

**Table 3 pharmaceutics-12-00509-t003:** The effect of artemisone in micro-emulsion on schistosomal egg production and liver granuloma load.

Treatment	# Mouse	Eggs/cm^2^	Granulomas/cm^2^
SMEDDS-20	1	47	41
	2	72	20
	3	172	76
	4	63	68
	5	70	50
SMEDDS-20-ART	1–6	0	0

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
