# Peer review of "Oral Administration of Artemisone for the Treatment of Schistosomiasis: Formulation Challenges and In Vivo Efficacy"

_pharmaceutics, 2020, doi:10.3390/pharmaceutics12060509_

Round 1

Reviewer 1 Report

In this paper by Zech and colleagues, the authors describe the development of a self-microemulsifying drug delivery system (SMEDDS) for the solubilization of the artemisone drug for the treatment of schistosomiasis as well as for other parasitic infections. After having confirmed the solubility, storage and physiochemical properties of the developed formulation, the authors assessed its efficacy in vivo using a mouse model of infection with S. mansoni. Notwithstanding this paper can be of some interest to the scientific community working in the field, I think that before considering it for publication in Pharmaceutics, the authors should address the following criticisms.

  • There are some typos in the whole manuscript. I would suggest the authors to carefully review and amend it accordingly.

  • It is not clear if the SMEDD is approved for human use or if there are any references or previous works supporting the use of SMEDDS in humans or its possible approval for human administration?

  • Did the authors test the in vivo efficacy of SMEDD-formulated artemisone prepared at different time points in order to assess if the storage of artemisone is affected over time?

Reviewer 2 Report

Comments on the  Manuscript ID: pharmaceutics-793400

Title: Oral administration of artemisone against schistosomiasis: formulation challenges and in vivo efficacy.’

General comments

The work presents an investigation of in vivo studies on mice infected with S. Mansoni using an optimized SMEDDS formulation. The formulation and in vivo aspects are very well covered. A more concise and focused introduction and a better organized presentation of the formulation development with  separation under suitable titles the applied physicochemical, technological and stability tests will make easier the reading.

Below are some points that authors may find useful for improvement of the manuscript.

Specific points

L 24 – Please change transformation to translation

L 72-79 - Reconsider presentation of this paragraph. It is not clear whether subcutaneous or oral route is preferred

Line 73 - correct means

There are contradicting statements, line  by replacing oral treatment ... with subcutaneous injection,    then  agian would be prepferable to adminsiter the drug orally

L 98 - Please add reference after artemisone

L 101,102 – ‘… or a formulation with low water activity would be desirable to achieve sufficient stability ..’. It is not clear what these concentrations (‘water activity’) in water are.

 L 107-111.Text between these line could either be removed or transferred to the beginning of the introduction

L 113, 120 – All this detailed account disadvantages is not necessary as it dilutes the objectives of the study given below.

L 126 129 – The reader expects to see the objective here not the desired properties

L 144 – 150 – HLB of surfactant for a type III SMEDDS is important. Some reference on the range of HLB for such SMEDDS is needed.

L 147 – In this work ternary phase diagrams are used for screening not for optimization.For the latter authors have performed further tests.

L168 - Did HS 15 melting not give clear liquid? why homogenization was needed?

L  199 Please change rates with percentages

L 206 – Please delete ‘with the exception of water’

L 212 – Which are these components?

L 232  - please mention the intervals

L 235 – please change against with ‘at different’

L 247 Add some info the onte measurement of density

L 250 cone-plate viscometer for liquids is impossible

L 279 – please add worms after adults

L 294 – please explain NIH

L 298 – Please explain how these cercariae were further utilized.

L 310 - This section can be split into parts which can be transferred to paragraphs of the measurement of the biologic activity above

Figure – Did authors not observed formation of liquid with milky appearance?

L 352 – Fig 2 does not show any such system

L 359 – This sentence is wrong. Please rephrase making clear that it refers to the speed of self-emulsification

Table 1 – Very high levels of surfactants plus propylene glycol raise thoughts about interference with function of natural body digestions mechanisms and possible toxicity issues. Authors should add some comment on this    

L 395 – 425 – This part can be presented as a separate section on Physical stability

L 420 – Is this rheometer different from that described in the methods?

L 573, 574 – Please change middle to medium

Round 2

Reviewer 1 Report

The authors addressed my criticisms.